# Distinct Signaling Pathways for Autophagy-Driven Cell Death and Survival in Adult Hippocampal Neural Stem Cells

**DOI:** 10.3390/ijms24098289

**Published:** 2023-05-05

**Authors:** Seol-Hwa Jeong, Hyun-Kyu An, Shinwon Ha, Seong-Woon Yu

**Affiliations:** Department of Brain Sciences, Daegu Gyeongbuk Institute of Science and Technology (DGIST), Daegu 42988, Republic of Korea; seolhwa@dgist.ac.kr (S.-H.J.); hkan@dgist.ac.kr (H.-K.A.); sh12@jhmi.edu (S.H.)

**Keywords:** autophagy, autophagic cell death, ERK, JNK, GSK-3β, adult hippocampal neural stem cells

## Abstract

Autophagy is a cellular catabolic process that degrades and recycles cellular materials. Autophagy is considered to be beneficial to the cell and organism by preventing the accumulation of toxic protein aggregates, removing damaged organelles, and providing bioenergetic substrates that are necessary for survival. However, autophagy can also cause cell death depending on cellular contexts. Yet, little is known about the signaling pathways that differentially regulate the opposite outcomes of autophagy. We have previously reported that insulin withdrawal (IW) or corticosterone (CORT) induces autophagic cell death (ACD) in adult hippocampal neural stem (HCN) cells. On the other hand, metabolic stresses caused by 2-deoxy-D-glucose (2DG) and glucose-low (GL) induce autophagy without death in HCN cells. Rather, we found that 2DG-induced autophagy was cytoprotective. By comparing IW and CORT conditions with 2DG treatment, we revealed that ERK and JNK are involved with 2DG-induced protective autophagy, whereas GSK-3β regulates death-inducing autophagy. These data suggest that cell death and survival-promoting autophagy undergo differential regulation with distinct signaling pathways in HCN cells.

## 1. Introduction

Macroautophagy (herein referred to as autophagy) is an evolutionarily conserved process that plays an essential role in development, innate immune defense, tumor suppression, and cell survival [1,2,3]. Autophagy fulfills these roles through the lysosomal degradation of harmful dysfunctional organelles, unneeded proteins, and intracellular pathogens [4,5]. The proteins encoded by the autophagy-related (ATG) genes govern the entire autophagy process [6]. In addition, several key upstream regulators have been reported, including the mammalian target of rapamycin (mTOR), phosphoinositide-3 kinase (PI3K), mitogen-activated protein kinase (MAPK), and AMP-activated protein kinase (AMPK) [7,8,9,10,11].

At the basal level, autophagy occurs constitutively and thereby plays an essential role in the turnover of cytoplasmic materials, such as proteins and organelles, and their recycling [1,2,12]. Moreover, autophagy can be activated by cellular stressors (such as nutrient and energy starvation) [13] or growth factor deprivation, hypoxia, oxidative stress, and pathogen infection [14,15,16]. Autophagy is essential for providing amino acids and other biochemical constituents which can be used for energy production and viability under starvation conditions [1,17].

Although autophagy is considered a protective process following cellular stressors, including starvation, recently, its role in cell death has been documented. The blockage of caspase-8 via treatment of Z-VAD kills mouse L929 fibrosarcoma cells through autophagy machinery ATG7 and Beclin-1 [18]. The inhibition of caspase-10 activates autophagic cell death (ACD) through the cFLIP-BCLAF axis in myeloma [19]. In addition to the examples of ACD following apoptosis inhibition, ACD occurs in A549 lung cancer cells accompanied by an autophagy flux increase, activating GBA1, a Gaucher-disease-associated gene [20]. In the human MCF7 breast cancer cell line, silibinin triggers cell death induced by autophagy, accompanied by mitochondrial malfunction [21]. For therapeutic trials, recent studies have focused on ACD induction in cancer cells that have defects in apoptosis, i.e., neferine [22], plasma-activated medium (PAM) [23], larotrectinib [24], and snake venom [25]. 

We have previously reported that insulin withdrawal (IW) or corticosterone (CORT) treatment caused ACD in adult hippocampal neural stem (HCN) cells [26,27,28]. In addition, several genetic molecules have been demonstrated to regulate ACD, such as glycogen synthase kinase-3beta (GSK-3β), calpain, AMPK, valosin-containing protein, ryanodine receptor 3, and Parkin [29,30,31,32,33,34]. HCN cells are regarded as a genuine model for understanding ACD mechanisms because IW- and CORT-induced autophagy fulfills the criteria for ACD in apoptosis-intact cells [26,27,35]. However, it remains a conundrum how autophagy can yield opposite outcomes; survival **vs.** death.

Here, we aimed to examine the distinct signaling pathways underlying two different roles of autophagy by comparing autophagy-driven survival and death conditions in HCN cells. For the induction of survival-promoting autophagy, we employed 2-deoxy-D-glucose (2DG). 2DG is a non-metabolizable glucose analog that blocks hexokinase, the rate-limiting enzyme of glycolysis [36,37]. When glycolysis is inhibited, energy depletion is expected, leading to autophagy activation [38]. To our knowledge, there has been no report that systemically compares signal transduction pathways depending on the different roles of autophagy in one cell type. Our data will contribute to understanding the sophisticated mechanisms that regulate autophagy in cell survival and death. 

## 2. Results

### 2.1. Four Cellular Stress Conditions Trigger Autophagy in HCN Cells

To examine the signaling pathways related to the contradictory roles of autophagy, we exposed HCN cells to three conditions that induce autophagy: IW, CORT for cell death and 2DG, and GL for cell survival. To validate autophagy induction, we checked a marker of the autophagosome formation, microtubule-associated protein 1A/1B-light chain 3 (LC3)-II, a lipidation form of LC3 [39]. In HCN cells, these treatments caused a marked increase in autophagic flux, as evidenced by the induction of LC3-II protein levels (Figure 1A–D). To confirm an increase in the autophagic flux, we used bafilomycin A1 (BafA_1_), an inhibitor of lysosome-autophagosome fusion [4]. Further increases in LC3-II levels by BafA_1_ indicate that the accumulation of LC3-II by three treatments is not due to impaired autophagy but the induction of autophagy flux (Figure 1A–D). In addition to BafA_1_ treatments, we used the mRFP-GFP-LC3 system to verify autophagic flux induction. Since the fluorescent signal of the GFP is quenched in the acidic autolysosomal environment, the RFP-only (red) signal represents the autolysosome, while the presence of both GFP and RFP (yellow) show autophagosomes [40]. Given the advantage of mRFP-GFP-LC3 characteristics, we measured autophagosome and autolysosome formations and observed increases in the total numbers of autophagic vesicles and autolysosomes, indicating the successful maturation of autophagosomes to autolysosomes (Figure 1E,F). These data show that three different stress conditions trigger autophagy in HCN cells.

### 2.2. 2DG and GL Did Not Affect Cell Viability, but IW and CORT Induced Cell Death Which Met Autophagic Cell Death Criteria

We previously reported that HCN cells undergo ACD following IW and CORT treatments [26,27,28]. In agreement with our previous reports, IW and CORT showed progressively enhanced cell death (Figure 2A). However, 2DG treatment and GL hardly affected cell viability until 48 h (Figure 2A). There was no activation of caspase-3, as shown by no cleaved form of caspase-3 (c-CASP3) and PARP-1 cleavage by IW or CORT, compared to staurosporine (STS) treatment, which was utilized as a positive control of apoptosis (Figure 2B). To show the causative role of autophagy in IW- or CORT-triggered cell death, we used the CRISPR-Cas9 gene editing technique for ULK/Atg1 (unc-51-like autophagy activating kinase 1) knockout (sg*Ulk1*) and lentiviral shRNA for Atg7 knockdown (sh*Atg7*). As expected, *Ulk1* or *Atg7* deficiency remarkably abrogated the cell death rate in 48 h, indicating that autophagy was causative to cell death following IW or CORT treatment in HCN cells (Figure 2D,H). Furthermore, Western blotting analyses of LC3-II levels in combination with BafA_1_ show that autophagic flux was reduced in sg*Ulk1* and sh*Atg7* HCN cells after IW and CORT treatments (Figure 2E,F,I,J). These data again confirm that the IW- and CORT-induced death of HCN cells meets all the criteria of ‘ACD’.

### 2.3. 2DG-Induced Autophagy Is Cytoprotective and Reduces IW- or CORT-Induced Cell Death

In HCN cells, 2DG-stimulated autophagy did not lead to cell death until 48 h (Figure 2A). To test whether 2DG-induced autophagy was protective (Figure 3A), we exposed HCN cells to 2DG prior to IW or CORT (Figure 3B). Interestingly, 2DG pretreatment prevented IW- or CORT-induced cell death compared to IW or CORT only (Figure 3C). Based on these pretreatment results, we suggest 2DG-induced autophagy acts as a protective autophagy in HCN cells.

### 2.4. The Inhibition of MAPK Blocks 2DG-Induced Autophagy but Not IW- or CORT-Induced Autophagy

Metabolic stress was reported to induce autophagy via MAPK [41,42]. Therefore, we examined whether the MAPK pathway is involved in 2DG-induced autophagy in HCN cells. MAPK inhibition was carried out using PD98059 as an ERK inhibitor (ERKi), SP600125 as a JNK inhibitor (JNKi), and SB202190 as a p38 inhibitor (p38i). Interestingly, all three inhibitors enhanced the cell death rate in 2DG (Figure 4A) and decreased autophagy, as shown by immunoblotting analyses of LC3-II levels (Figure 4B,C), indicating the involvement of MAPK in 2DG-induced survival autophagy in HCN cells. However, MAPK inhibition did not affect cell death and autophagic flux after IW and CORT treatment (Figure 5A–F), suggesting that the MAPK pathway is not involved in death-inducing autophagy.

### 2.5. The Blockage of ERK and JNK Impedes the Protection Effects of 2DG

We previously determined 2DG as a protective autophagy-inducing condition, as suggested by decreases in IW- or CORT-induced cell death by 2DG pretreatment (Figure 3C). Since MAPK inhibition subjugated 2DG-induced autophagy, we speculated whether MAPK inhibition would abolish the protective action of 2DG against IW or CORT. Although p38i reduced 2DG-induced autophagic flux, its effect was less potent than ERKi or JNKi (Figure 4B,C). Therefore, we only tested ERKi and JNKi. While the pretreatment of 2DG effectively reduced IW- or CORT-induced cell death, pretreatment with 2DG coincubated with ERKi or JNKi did not decrease cell death in HCN cells (Figure 4D). In summary, our results suggest that ERK and JNK are required for 2DG-induced autophagy, which has a protective role in HCN cells.

### 2.6. The Inhibition of GSK-3β Blocks IW- or CORT-Induced ACD but Not 2DG-Induced Autophagy

We previously reported the pivotal role of GSK-3β in mediating IW-induced ACD in HCN cells [30]. Because CORT also showed ACD characteristics, we tested the involvement of GSK-3β in CORT-induced ACD in HCN cells. BIO, a specific GSK-3β inhibitor, suppressed IW-induced cell death [30]. As expected, BIO also reduced CORT-triggered cell death (Figure 6A). A decrease in the cell death rate due to BIO was significant, but not remarkable. It is plausible that other proteins previously reported as the regulators of IW- or CORT-induced ACD, such as calpain 2, AMPK, valosin-containing protein, ryanodine receptor 3, and parkin, may still be active in BIO-treated HCN cells [29,31,32,33,34]. BIO also mitigated IW- or CORT-triggered autophagic flux (Figure 6B–E), indicating the critical role of GSK-3β in the CORT-induced as well as IW-induced ACD in HCN cells. On the contrary, BIO treatment did not affect either cell death or autophagy flux in 2DG-treated HCN cells (Figure 6E,F). These results reveal the distinct signaling pathways between protective and death-inducing autophagy in the same cell type with MAPK for 2DG-induced autophagy and GSK-3β for IW- or CORT-induced ACD. 

### 2.7. IW and CORT, but Not 2DG or GL, Reduce ATP Amounts and Increase Mitochondrial ROS Levels in HCN Cells

Because mitochondria generate 90% of the cellular energy, insufficient mitochondrial energy production may elicit cell death [43]. We also reported that mitophagy is induced via Parkin after IW in HCN cells [29,33]. To investigate the critical differences between death and survival autophagy, we measured intracellular ATP contents and found a significant decline in ATP in HCN cells after IW or CORT treatment for 12 h. On the other hand, ATP levels were well maintained in 2DG or GL-treated HCN cells (Figure 7A). Survival autophagy may provide energy via the recycling of intracellular materials.

Moreover, mitochondria generate massive ROS as a by-product in response to stress [43]. We analyzed mitochondrial ROS levels and found considerable ROS accumulation after IW or CORT treatment for 24 h, while 2DG or GL did not induce mitochondrial ROS production (Figure 7B,C). Altogether, we observed that IW- or CORT-induced ACD was accompanied by significant ATP depletion and mitochondrial ROS accumulation.

## 3. Discussion

Autophagy pathways have been extensively studied in a variety of cellular settings, including cell survival and death. With regards to cell survival and death in particular, strict criteria of ACD were proposed [35]: (1) other types of programmed cell death are not involved; (2) both autophagy markers and autophagy flux should be increased; and (3) the inhibition of autophagy with pharmacological or genetic methods prevents cell death [35]. Nevertheless, many cases of ACD have not fully elucidated the underlying signaling mechanisms. So, it is poorly understood how the same autophagy process can lead to different endings of its action. Here, we employed three autophagy-activating conditions and observed protective autophagy via 2DG and death-inducing autophagy via IW and CORT in one cell type. By using pharmacological inhibitors, we showed that MAPK and especially ERK and JNK are related to 2DG-triggered autophagy, while GSK-3β regulates death-driving autophagy. These results prove that a different signaling mechanism could regulate two different autophagy abilities. 

So, what would selectively trigger these kinases? Apoptosis is ATP-dependent cell death and favored by increased cytosolic ATP [44,45,46]. Based on the study by Shimizu et al., intracellular ATP levels are sustained until the mitochondria gradually lose their membrane potential and stop ATP production [47]. On the other hand, the early abrupt aberration of mitochondria that fails to produce ATP may be the starting point of ACD since the impaired dimerization of ATP synthase showed autophagy-dependent cell death in *Podospora anserine* [48]. Therefore, it is possible that a sharp decrease in mitochondrial function might be responsible for autophagy, partly through the promotion of mitophagy to cause cell death [33]. In contrast, the maintenance of ATP levels may lead autophagy to protection or later apoptosis after the gradual depletion of cellular energy. We suggest that the abrupt malfunction of mitochondria, such as a rapid reduction in ATP production, may be one of key traits of ACD.

Based on the above notion, different mitochondrial functional statuses may also trigger different upstream signaling kinases, thereby determining the role of autophagy between survival and death. Different amounts or production kinetics of mitochondrial reactive oxygen species could be a decisive factor, which may determine the ATP depletion rate.

In summary, we elucidated the existence of survival autophagy and death-inducing autophagy in HCN cells following different autophagy induction conditions. Our results suggest that ERK and JNK are crucial pathways in 2DG-induced protective autophagy. Conversely, GSK-3β mediates ACD in HCN cells. Further studies on the molecular mechanism and relationship with functions of subcellular organelles, including mitochondria, will be required to advance our knowledge on the differential regulation of autophagy between survival and death.

## 4. Materials and Methods

### 4.1. Materials

The following reagents were used BafA_1_ (BML-CM110-0100) from Enzo life Sciences; BIO (B1686) from Sigma-Aldrich, St. Louis, MO, USA; ERKi (9900), JNKi (8177), p38i (8158), and STS (9953) from Cell Signaling Technology (Danvers, MA, USA). Antibodies were used against the following proteins: LC3B (NB100-2220) from Novus Biologicals (Littleton, CO, USA); ULK1 (8054), ATG7 (8558), CASP3 (9661), and PARP-1 (9542) from Cell Signaling Technology; and horseradish peroxidase-conjugated ACTB (sc-4778) from Santa Cruz Biotechnology (Dallas, TX, USA).

### 4.2. Cell Cultures

HCN cells were obtained from the hippocampus of 8-week-old Sprague Dawley rats and maintained in coated culture dishes with poly-L-ornithine and laminin, as previously described [26]. HCN cells were cultured in a chemically defined serum-free media containing Dulbecco’s modified Eagle’s medium (DMEM)/F-12 (12400–024, Thermo Fisher Scientific, Waltham, MA, USA), supplemented with individually prepared N2 components and basic fibroblast growth factor (bFGF) (20 ng/mL; 100-18B-500, Peprotech, Cranbury, NJ, USA). Insulin withdrawal media were prepared by omitting insulin. To make glucose-low media, Dulbecco’s modified Eagle’s medium (DMEM)/F-12 (D9807-02, USBiological, Salem, MA, USA) was supplemented with N2 components and bFGF. Then, D-glucose (Sigma-Aldrich, St. Louis, MO, USA) was added to a final concentration of 5 mM.

### 4.3. Cell Death Assay

In a 96-well plate, HCN cells were seeded to a coated plate at a density of 1.5 × 10^4^ cells per cm^2^. Seeded HCN cells were stained using Hoechst 33342 (P3566, Invitrogen,) and propidium iodide (PI) (P4170, Sigma-Aldrich). The PI solution (10 mg/mL) and Hoechst (1 mg/mL) were diluted at 1:2000 and incubated in the dark at 37 °C for 20 min. Under a fluorescence microscope (Axiovert 40 CFL, Carl Zeiss, Oberkochen, Germany), stained cells were imaged and analyzed using Matlab with CellC package analysis software. The rate of cell death was calculated as follows: PI-positive cell number/Hoechst-positive cell number × 100.

### 4.4. Generation of Stable Cell Lines

Short hairpin RNA (shRNA), delivered against rat *Atg7* (TRCN0000092164 and TRCN0000369085), and a nontargeting shRNA control were delivered with a lentivirus-expressing vector pLKO.1 (Sigma-Aldrich). Lentiviruses were produced following published protocols and were used to infect HCN cells [29,49]. Cells were selected with 10 μg/mL of puromycin. Knockdown efficiency was validated by the immunoblotting of ATG7. sg*Ulk1* knockout cells were generated using the CRISPR/Cas9 system with the guide RNA (gRNA) sequence for rat *Ulk1* (NCBI Gene ID: 360827). gRNA was designed and purchased from ToolGen (Republic of Korea), as previously described [50].

### 4.5. Transfection

ptfLC3 (encoding mRFP-GFP-LC3, 21074, deposited by Tamotsu Yoshimori) was lpurchased from Addgene. The transfection of plasmids was performed using Lipofectamine 2000, in accordance with the manufacturer’s instructions. After transfection for 4 h, the medium was removed and incubated for another 48 h with newly changed media. Transfected cells were then exposed to autophagy induction conditions and used for fluorescence imaging. 

### 4.6. Confocal Imaging

Cells were fixed with 4% paraformaldehyde (P6148, Sigma-Aldrich). The nuclei were stained with Hoechst 33342. Images were obtained using an LSM 800 confocal microscope (Carl Zeiss) with a 63 x oil objective lens and analyzed with ZEN software (Carl Zeiss). RFP signals and GFP signals were obtained by 561 nm and 488 nm filters, respectively.

### 4.7. Western Blotting

Harvested HCN cells were lysed in radioimmunoprecipitation assay buffer (Sigma-Aldrich, 89901) containing 1 × protease and phosphatase inhibitor cocktails (78444, Thermo Fisher Scientific, Waltham, MA, USA) for 30 min. Every 10 min, the lysates were vortexed. After centrifugation (16,000× *g*, 20 min), the supernatants were collected. Using a BCA protein assay reagent, the protein concentrations were measured using a BCA protein assay reagent (23225, Thermo Scientific). Typically, 5–10 µg of proteins per well were loaded in an SDS-polyacrylamide gel. After running on an SDS-polyacrylamide gel, proteins were transferred to a polyvinylidene fluoride membrane using a semi-dry electrophoretic transfer cell (BioRad, Hercules, CA, USA) and membranes were blocked for 30 min at room temperature with 5% nonfat dry milk. The primary antibodies were prepared according to the manufacturers’ recommendations and the membranes were incubated with primary antibodies overnight. After washing three times for 10 min each, the membrane was exposed for 1 h with peroxidase-conjugated secondary antibodies diluted in 2.5% blocking solution. The membranes were then processed using a chemiluminescence detection kit (34580, Thermo Scientific). All Western blots were quantified using ImageJ (NIH) software (version 1.54d) and normalized to ACTB.

### 4.8. MitoSOX Staining Using Fluorescence-Activated Cell Sorting (FACS) Analysis

HCN cells were stained with MitoSOX Red, a mitochondrial superoxide indicator (M36007, Invitrogen), for 20 min in the dark. Cells were harvested using 0.05% trypsin (SH30236.02, HyClone, Logan, UT, USA) and centrifuged at 1000× *g* for 3 min. Cell pellets were resuspended in PBS. All stained cells were analyzed with an Accuri C6 flow cytometer (BD Biosciences, Franklin Lakes, NJ, USA).

### 4.9. ATP Measurement Assay

A CellTiter-Glo 2.0 kit (G9241, Promega, Madison, WI, USA) was used in line with the manufacturer’s instructions. HCN cells were plated at a density of 2 × 10^4^ cells per well onto 96-well white plates. The plates were coated with poly-L-ornithine and laminin. After the reagent solutions were treated to each well, the plate was incubated for 20 min on an orbital shaker at room temperature. Luminescence was detected using a SpectraMax L (Molecular Devices, San Jose, CA, USA). We used ATP disodium salt to generate an ATP standard curve.

### 4.10. Statistical Analysis

All indicated values are presented as mean ± standard errors of the mean (SEM). Data were taken from at least three independent experiments. Statistical significance was validated by one-way analysis of variance (ANOVA), after post hoc Tukey’s test was used for multiple groups using GraphPad Prism (Version 8.0) (GraphPad Software, USA).

## Figures and Tables

**Figure 1 ijms-24-08289-f001:**
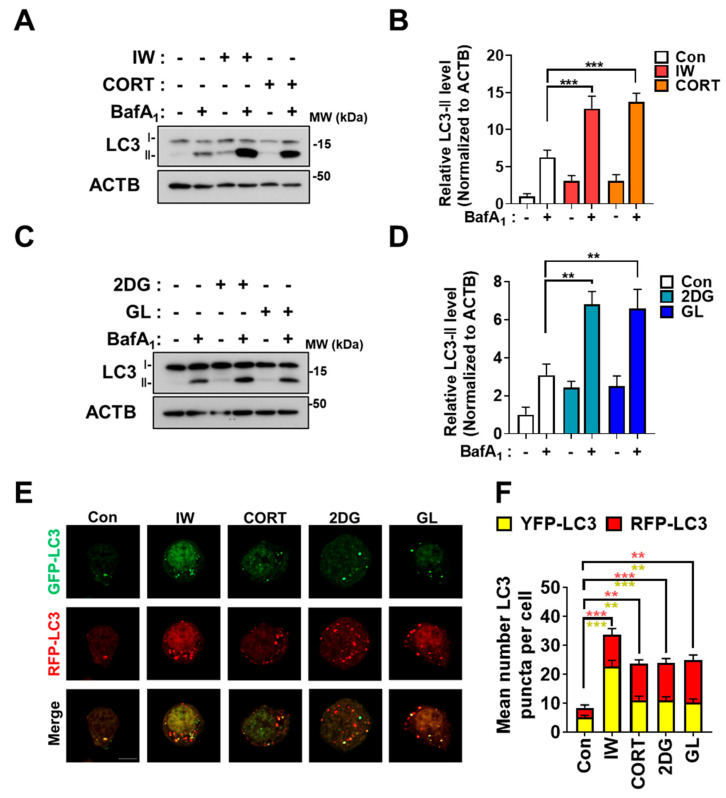
IW, CORT, 2DG, and GL increase the autophagic flux in HCN cells. (**A**) Western blotting analyses of LC3−II levels in HCN cells after IW for 24 h and CORT (200 μM) treatments for 48 h. Bafilomycin A1 (BafA_1_, 10 nM) was added 1 h before cell harvest. (**B**) The quantification of LC3−II levels after normalization to ACTB (n = 3). (**C**) Western blotting analyses of LC3−II levels after 2DG treatment (250 μM) and GL for 24 h. BafA_1_ (10 nM) was added for 1 h before the cell harvest. (**D**) The quantification of LC3−II levels after normalization to ACTB (n = 3). (**E**) The measurement of autophagic flux using the mRFP−GFP−LC3 puncta assay. HCN cells were treated under each condition for 6 h. (**F**) The quantification of mRFP−GFP−LC3B puncta (n = 18 cells from 5 experiments per each condition). + in the Western blots indicates the treatment condition. Scale bar, 5 µm. Con, control. ** *p* < 0.01 and *** *p* < 0.001 for the indicated comparisons.

**Figure 2 ijms-24-08289-f002:**
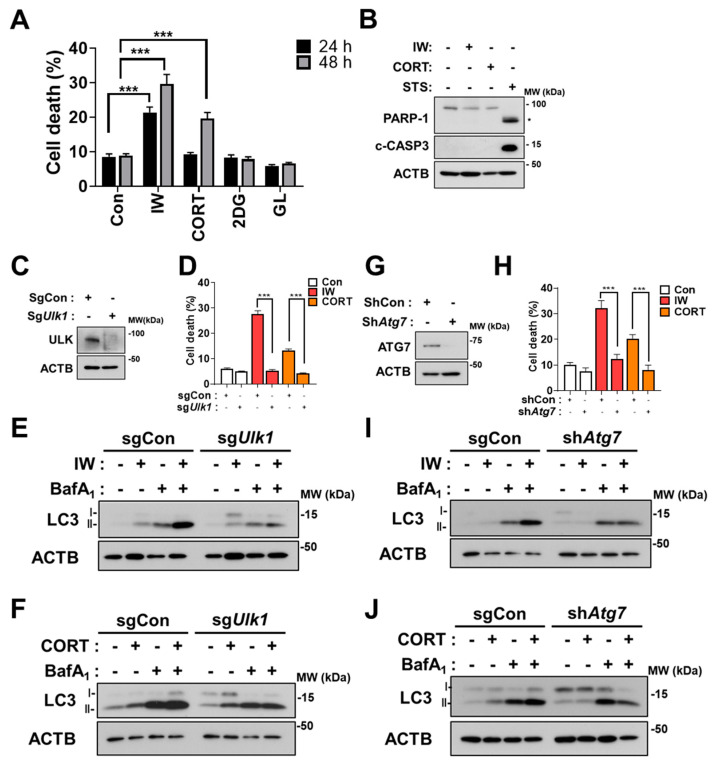
IW and CORT, but not 2DG and GL, induce cell death in HCN cells. (**A**) Cell death rates in HCN cells after IW, CORT, 2DG, and GL treatment for 24 h and 48 h (n = 3). (**B**) Western blotting analyses of the cleaved forms of caspase 3 (c−CASP3) and PARP−1 after treatment with the indicated conditions for 24 h. * PARP-1 cleaved fragment. Staurosporine (STS, 0.5 μM) was added for 6 h as a positive control for apoptosis. (**C**) The validation of ULK1 knockout by Western blotting analysis in HCN cells. (**D**) Cell death rates in sgCon and sg*Ulk1* HCN cells after IW and CORT for 48 h (n = 3). (**E**,**F**) Western blotting analyses of LC3−II levels after IW (24 h) and CORT (48 h) treatment in sgCon and sg*Ulk1* HCN cells. BafA_1_ (10 nM) was added for 1 h before cell harvest (n = 3). (**G**) The validation of ATG7 knockdown using Western blotting analysis in HCN cells. (**H**) Cell death rates in shCon and sh*Atg7* HCN cells after IW and CORT treatment for 48 h (n = 3). (**I**,**J**) Western blotting analyses of LC3−II levels after IW (24 h) and CORT (48 h) treatment in shCon and sh*Atg7* HCN cells. BafA_1_ (10 nM) was added for 1 h before cell harvest. The blots were representative of three experiments with similar results. + in the Western blots indicates the treatment condition. Con, control. *** *p* < 0.001 for the indicated comparisons.

**Figure 3 ijms-24-08289-f003:**
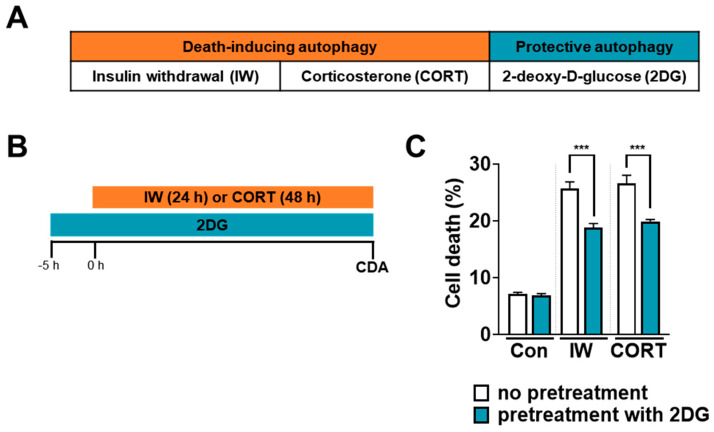
2DG-induced autophagy prevents IW− or CORT−induced cell death in HCN cells. (**A**) The classification of IW− or CORT−induced autophagy as death-driving autophagy and 2DG−induced autophagy as survival−driving autophagy. (**B**) A schematic diagram for the pretreatment of 2DG prior to IW or CORT treatment. (**C**) Decreases in cell death rates via the pretreatment of 2DG for 5 h before IW (24 h) or CORT (48 h) treatment (n = 3). Con, control; CDA, cell death assay. *** *p* < 0.001 for the indicated comparisons.

**Figure 4 ijms-24-08289-f004:**
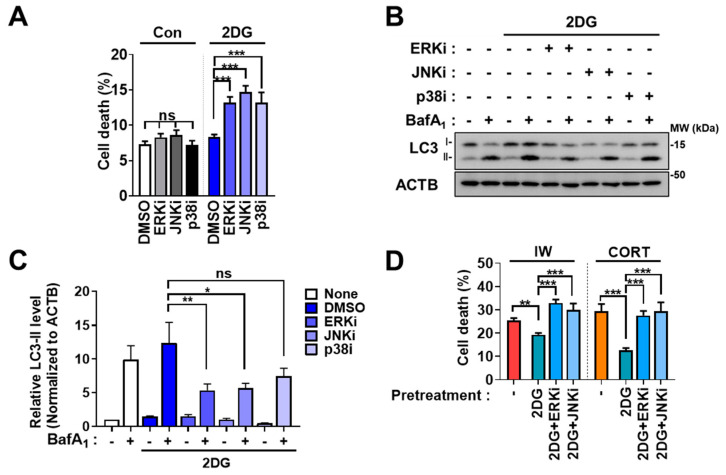
MAPK is involved in 2DG−induced autophagy in HCN cells. (**A**) The effects of MAPK inhibitors on cell death after 2DG treatment for 24 h (n = 3). ERKi (15 µM), JNKi (2.5 µM), and p38i (1 µM) were treated. (**B**) Analyses of the effects of MAPK inhibition on autophagic flux using Western blotting analyses of LC3−II levels after 2DG treatment for 24 h. (**C**) The quantification of LC3−II levels after normalization to ACTB (n = 3). (**D**) The effects of ERKi and JNKi on the protective action of 2DG pretreatment against IW (24 h) or CORT (48 h) treatment (n = 3). + in the Western blots indicates the treatment condition. Con, control; ERKi, ERK inhibitor; JNKi, JNK inhibitor; p38i, p38 inhibitor. * *p* < 0.05, ** *p* < 0.01, *** *p* < 0.001; ns, not significant for the indicated comparisons.

**Figure 5 ijms-24-08289-f005:**
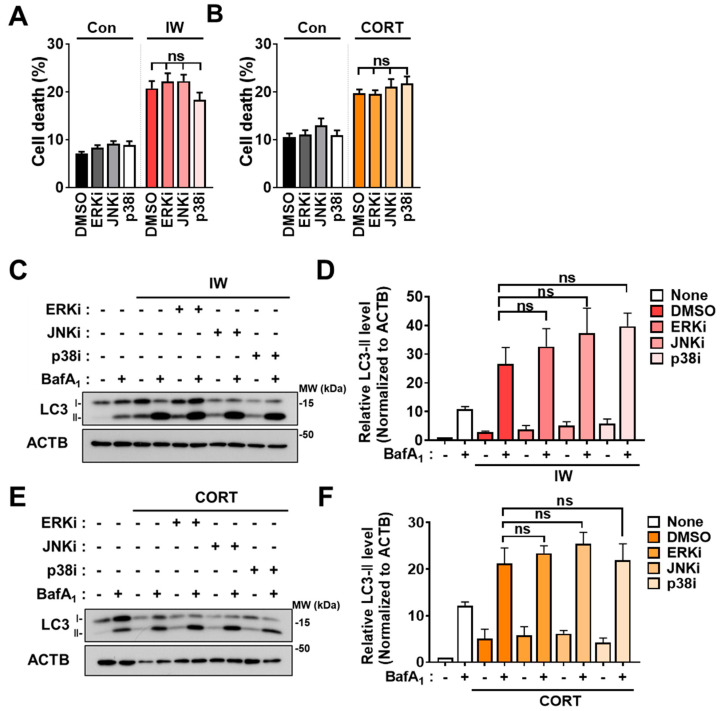
MAPK inhibitors did not alter autophagic flux and death rates following IW and CORT. (**A**) Cell death rates of MAPK inhibitors treated with IW for 24 h (n = 8). ERKi (15 µM), JNKi (2.5 µM), and p38i (1 µM) were treated. (**B**) Cell death rates of MAPK inhibitors added with CORT for 48 h (n = 4). ERKi (7.5µM), JNKi (1 µM), and p38i (0.5 µM) were treated. (**C**) Western blotting analyses of LC3B following IW with ERKi and JNKi. The blots shown are representative of 3 independent experiments with similar results. (**D**) The quantification of LC3B−II after normalization to ACTB of (**C**) (n = 3). (**E**) Western blotting analyses of LC3B when CORT with ERKi and JNKi. The blots shown are representative of 3 independent experiments with similar results (n = 3). (**F**) The quantification of LC3B−II after normalization to ACTB of (**E**) (n = 3). + in the Western blots indicates the treatment condition. Con, control; ERKi, ERK inhibitor; JNKi, JNK inhibitor; p38i, p38 inhibitor; ns, not significant for the indicated comparisons.

**Figure 6 ijms-24-08289-f006:**
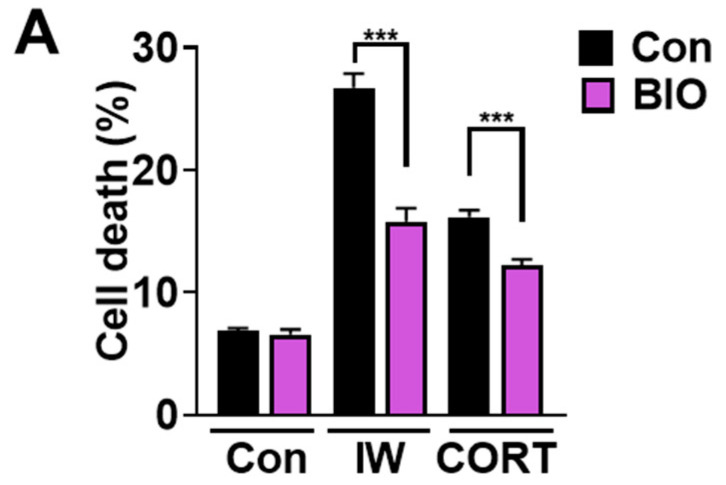
GSK−3β is involved in IW− or CORT−induced ACD, but not 2DG-induced autophagy. (**A**) The effects of GSK−3β inhibitor, BIO (0.25 µM), on IW− or CORT−induced cell death. HCN cells were treated with IW and CORT for 48 h with or without BIO (n = 5). (**B**,**D**,**F**) Western blotting analyses of LC3B−II levels after IW (24 h), CORT (48 h), and 2DG (24 h). (**C**,**E**,**G**) The quantification of LC3B−II levels after normalization to ACTB of (**C**) (n = 3), (**D**) (n = 4), and (**E**) (n = 5). + in the Western blots indicates the treatment condition. Con, control. ** *p* < 0.01; *** *p* < 0.001; ns, not significant for the indicated comparisons.

**Figure 7 ijms-24-08289-f007:**
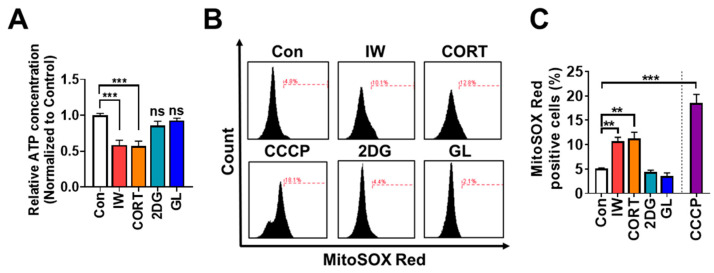
IW and CORT, but not 2DG or GL, induce ATP reduction and ROS accumulation. (**A**) ATP measurement assay after IW, CORT, 2DG, and GL for 12 h. (n = 5) (**B**,**C**) FACS analysis of mitochondrial ROS using MitoSOX Red (100 nM) after IW, CORT, 2DG, GL, and carbonyl cyanide 3-chlorophenylhydrazone (CCCP) for 24 h. CCCP (10 µM) was treated as a positive control for 30 min before detection (n = 4). ** *p* < 0.01 and *** *p* < 0.001. ns, not significant for the indicated comparisons.

## Data Availability

Not applicable.

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
