# Peer review of "Distinct Signaling Pathways for Autophagy-Driven Cell Death and Survival in Adult Hippocampal Neural Stem Cells"

_ijms, 2023, doi:10.3390/ijms24098289_

Round 1

Reviewer 1 Report

Seol-Hwa Jeong et al. reported there exist distinct signaling pathways for autophagy-driven cell death and survival in adult hippocampal neural stem cells. MAPKs (ERK and JNK) are involved in 2DG-induced protective autophagy, whereas GSK-3β regulates death-inducing autophagy. The authors suggest that the different mitochondrial functional statuses may trigger different upstream signaling kinases, thereby determining autophagy’s roles between survival and death. The findings are interesting and important. However, the reviewer has some concerns.

1. The autophagy induction by 2-DG (in Figure 1C, D; Figure 4B, C ; Figure 6 F, G) is not remarkable.

2. A second metabolic stress is better employed to further strengthen the conclusions of this paper.

3.The kinetics of the production of ATP and mitochondrial reactive oxygen species can strongly support the conclusion and discussion, which is not provided in this study.

4. GSK-3β inhibitor, BIO, only partially downregulates IW or CORT-induced cell death (Figure 6A). It needs to be discussed.

The quality of English language is fine.

Reviewer 2 Report

The experimental work of S.-H. Jeong et al is devoted to the elucidation of a very topical question: why stimulation of autophagy causes cell death in some cases and promotes cell survival in others.

Using hippocampal HCN cells as the subject, the authors showed that insulin withdrawal (IW) and corticosterone (CORT), but not 2-deoxy-D-glucose induce cell death. All these three effectors induce autophagy, which is MAPK (ERK-, JNK-, p38-)-dependent (2DG) and GSK-3b-dependent (IW, CORT). DG prevents IW- and CORT- induced cell death, and this effect may be associated with induction of autophagy, which, however, may have its own specificity. According to the reviewer, the authors have conducted an interesting study on a very relevant topic. The reviewer has no major notes on the work.

Questions to authors.

1. Authors demonstrate that 2DG reduces cell death caused by IW and CORT treatment (Fig 3), in parallel with inducing autophagy (Fig 1). However, are these effects interrelated? Is this still the question? A possibility exists that different pathways (by ERK/JNK/p38 MAPKs) mediate these effects. What the authors think about it?

2. If autophagy is involved in these effects after all, what are the fundamental differences between this process in the case of cell death and the enhancement of cell viability? In the discussion, the authors talk about the participation of ATP and mitochondria in these processes. May be we are talking about macroautophagy and mitophagy (or some other selective autophagy). It would be desirable to discuss these two issues in the work

Minor remarks

Line 104: References 2-4 are incorrect and need to be corrected. Authors are encouraged to verify the reference list.

Line 108: Figure 2B instead of Figure 2A

Reviewer 3 Report

  This work is an interesting and well-organized study on a topical and important topic in the field of cell biology. The summary of this work is well structured, specific, contains objective information about the study and describes the problem as a whole. The introduction section briefly but succinctly describes the involvement of the autophagy process in various cellular model systems and justifies the need to study the different signaling pathways underlying the two different roles of autophagy by comparing the survival and death conditions mediated by autophagy in cells. The Results section is presented in a witty, concise manner and clearly demonstrates the results of the experiments.

However, there are comments to this section, I would like the authors not only to present the results in a concise form, but also to explain their design in more detail, in particular, it is necessary to explain in detail the meaning of + and - in the upper part of the diagrams in Fig. 1A, 1C, 2B, 2C, 2G, 2E-2J, 4B, 5C, 5E located above the immunoblots. In general, the results section needs more detail and emphasis to enable readers to better navigate the vast and complex material.

In the Discussion section, the authors briefly describe and discuss their results, focusing on putative signaling mechanisms that regulate ACD. It is necessary to appreciate the short and systematic form of this section, which not only analyzes two different abilities for autophagy, but also puts forward a reasonable hypothesis explaining these mechanisms based on the data obtained. Thus, this work is certainly commendable, although I believe that additional citation of ultrastructural data showing the flow of autophagy in cells, the state of mitochondria, and the presence or absence of autophagosomes, could provide direct evidence of this process in cell culture. Data demonstrating apoptosis in cells could also provide direct evidence for the results of the autophagy process and/or its absence.

In the Materials and Methods section, it is necessary to provide complete protocols, in particular, scanning modes and data on the used filters for sections 4.6. Confocal imaging. Section 4.8. Statistical analysis It is necessary to specify the entire statistical algorithm, including a description of the sample size and criteria for comparison

  This work is an interesting and well-organized study on a topical and important topic in the field of cell biology. The summary of this work is well structured, specific, contains objective information about the study and describes the problem as a whole. The introduction section briefly but succinctly describes the involvement of the autophagy process in various cellular model systems and justifies the need to study the different signaling pathways underlying the two different roles of autophagy by comparing the survival and death conditions mediated by autophagy in cells. The Results section is presented in a witty, concise manner and clearly demonstrates the results of the experiments.

However, there are comments to this section, I would like the authors not only to present the results in a concise form, but also to explain their design in more detail, in particular, it is necessary to explain in detail the meaning of + and - in the upper part of the diagrams in Fig. 1A, 1C, 2B, 2C, 2G, 2E-2J, 4B, 5C, 5E located above the immunoblots. In general, the results section needs more detail and emphasis to enable readers to better navigate the vast and complex material.

In the Discussion section, the authors briefly describe and discuss their results, focusing on putative signaling mechanisms that regulate ACD. It is necessary to appreciate the short and systematic form of this section, which not only analyzes two different abilities for autophagy, but also puts forward a reasonable hypothesis explaining these mechanisms based on the data obtained. Thus, this work is certainly commendable, although I believe that additional citation of ultrastructural data showing the flow of autophagy in cells, the state of mitochondria, and the presence or absence of autophagosomes, could provide direct evidence of this process in cell culture. Data demonstrating apoptosis in cells could also provide direct evidence for the results of the autophagy process and/or its absence.

In the Materials and Methods section, it is necessary to provide complete protocols, in particular, scanning modes and data on the used filters for sections 4.6. Confocal imaging. Section 4.8. Statistical analysis It is necessary to specify the entire statistical algorithm, including a description of the sample size and criteria for comparison

Round 2

Reviewer 1 Report

The manuscript has been much improved. This article can be accepted.